# A Linkage-Based Genome Assembly for the Mosquito *Aedes albopictus* and Identification of Chromosomal Regions Affecting Diapause

**DOI:** 10.3390/insects12020167

**Published:** 2021-02-16

**Authors:** John H. Boyle, Pasi M. A. Rastas, Xin Huang, Austin G. Garner, Indra Vythilingam, Peter A. Armbruster

**Affiliations:** 1Department of Biology, Georgetown University, 37th and O St, Washington, DC 20057, USA; john.h.boyle@gmail.com (J.H.B.); huangx87@upenn.edu (X.H.); aggarner@g.harvard.edu (A.G.G.); 2Department of Biology, University of Mary, Bismarck, ND 58504, USA; 3Institute of Biotechnology, Helsinki Institute of Life Science (HiLIFE), University of Helsinki, 00014 Helsinki, Finland; pasi.rastas@gmail.com; 4Department of Parasitology, Faculty of Medicine, University of Malaya, 50603 Kuala Lumpur, Malaysia; indrav@um.edu.my

**Keywords:** *Aedes albopictus*, mosquito genome, linkage map, diapause

## Abstract

**Simple Summary:**

A genome sequence can provide the basis for understanding a wide range of biological properties of an organism. For vector and pest species, this knowledge can be used to develop novel control strategies based on genome modifications that disrupt traits related to ecological adaptation and disease transmission. Here, we use a genetic mapping experiment to produce an improved version of a previous genome sequence for the invasive vector mosquito, *Aedes albopictus*. We then use the improved genome sequence to identify regions of the genome that contain candidate genes that may affect the ability of this mosquito to undergo overwintering dormancy, a crucial ecological adaptation.

**Abstract:**

The Asian tiger mosquito, *Aedes albopictus*, is an invasive vector mosquito of substantial public health concern. The large genome size (~1.19–1.28 Gb by cytofluorometric estimates), comprised of ~68% repetitive DNA sequences, has made it difficult to produce a high-quality genome assembly for this species. We constructed a high-density linkage map for *Ae. albopictus* based on 111,328 informative SNPs obtained by RNAseq. We then performed a linkage-map anchored reassembly of AalbF2, the genome assembly produced by Palatini et al. (2020). Our reassembled genome sequence, AalbF3, represents several improvements relative to AalbF2. First, the size of the AalbF3 assembly is 1.45 Gb, almost half the size of AalbF2. Furthermore, relative to AalbF2, AalbF3 contains a higher proportion of complete and single-copy BUSCO genes (84.3%) and a higher proportion of aligned RNAseq reads that map concordantly to a single location of the genome (46%). We demonstrate the utility of AalbF3 by using it as a reference for a bulk-segregant-based comparative genomics analysis that identifies chromosomal regions with clusters of candidate SNPs putatively associated with photoperiodic diapause, a crucial ecological adaptation underpinning the rapid range expansion and climatic adaptation of *A. albopictus*.

## 1. Introduction

Advances in high-throughput DNA-sequencing are driving rapid progress in genome analysis of a wide range of organisms, including many mosquitoes [1]. High-quality genome sequences can provide a basis for addressing a broad suite of fundamental questions concerning the evolutionary history, molecular physiology, and behavioral genetics of mosquito vectors [2,3,4,5]. Furthermore, integration of these genomic insights with emerging genetic engineering technologies presents a promising approach for developing novel methods to control vector abundance and pathogen transmission [6,7,8]. 

The mosquito genus *Aedes* contains many species capable of transmitting pathogens of veterinary and human significance [9]. Among the aedine species, *Aedes aegypti* and *Aedes albopictus* are the two most important vectors of human viruses, including dengue, chikungunya, and Zika; they are also the two aedine species with the most well-developed genomic resources. *Ae. aegypti* and *Ae. albopictus* are estimated to have diverged approximately 70 million years ago [10]. They appear to exhibit strong synteny based on comparative linkage mapping with RFLP markers [11], and both genomes are comprised of a large proportion of repetitive DNA elements, components of which are common to both species [10,12]. The percent identity of protein-coding gene sequences is also relatively high (~85%) based on BLAST alignments [13].

The sequencing and assembly of both the *Ae. aegypti* and *Ae. albopictus* genomes have been particularly challenging due to the large size and large repetitive DNA content of these genomes. Initial genome sequences for both species represented groundbreaking advances [10,14] but produced fragmented assemblies with relatively short scaffold sizes. This limitation was largely overcome for *Ae. aegypti* with a recent chromosome-level genome assembly (AaegL5.0) produced by sequencing 166 Gb with long-read PacBio technology, combined with 10X and Hi-C sequencing, leading to a 1.2 Gb genome assembly with 94% of sequence reads mapped to the three chromosomes [12]. Similarly, for *Ae. albopictus*, Palatini et al. (2020) also applied PacBio, 10X, and Hi-C sequencing to produce a substantially improved genome assembly, AalbF2. The AalbF2 assembly exhibits significantly improved contiguity (N50 scaffold length = 55.7 Mb) relative to the Chen et al. (2015) assembly, and 75% of the assembled genome maps to chromosomes. However, the AalbF2 assembly size is 2.54 Gb, which is approximately twice the expected size of 1.19–1.28 Gb based on cytofluorimetric estimates of genome size [15]. Identification of up to 1329 likely artifactually duplicated gene copies in the AalbF2 assembly further suggests the presence of haplotigs that were not able to be collapsed in the assembly [15].

Here, we construct a high-density linkage map for *Ae. albopictus* based on 111,328 informative single nucleotide polymorphisms (SNPs) obtained by RNAseq. We used RNAseq to identify SNPs because it is an efficient and affordable approach to obtain a reduced representation of the genome (i.e., transcribed regions). We then perform a linkage-map anchored reassembly of AalbF2. Anchoring a genome assembly on a linkage map is a powerful approach, especially when high-density linkage markers are available, because the map positions can be used to order contigs for scaffold assembly, as well as ordering scaffolds and removing haplotigs [16]. We refer to this new map-anchored reassembly as AalbF3 to denote that it is based on the contigs of Palatini et al. (2020). We demonstrate the utility of AalbF3 by using it as a reference for a bulk-segregant-based comparative genomics analysis that identifies chromosomal regions with clusters of candidate SNPs putatively associated with photoperiodic diapause. Although several previous studies have used RNAseq to identify genes affecting photoperiod diapause in *Ae. albopictus* [5,17,18,19,20], this is the first study to take a genetic mapping-based approach. Photoperiodic diapause is a form of dormancy that allows temperate-zone populations of *Ae. albopictus* to survive over winter. This trait is a crucial ecological adaptation underpinning rapid range expansion and climatic adaptation of *Ae. albopictus* across the temperate range of its global distribution [21], and a potential target for novel control strategies based on disrupting the diapause response.

## 2. Materials and Methods

### 2.1. Summary of Approach

Our study design is based on the observation that tropical populations of *Ae. albopictus* are genetically incapable of photoperiodic diapause and temperate populations have a robust, genetically determined diapause response [22,23]. Furthermore, preliminary data indicated that tropical and temperate populations differ at up to 1 million SNP sites across the genome. Our approach consists of the following five components illustrated in Figure 1: (1) We constructed a high-density linkage map by crossing a single tropical male to a single temperate female and genotyping 70 individuals at more than 111,328 informative SNP loci in the F_7_ intercross generation; (2) We used the resulting linkage map to create a chromosome-level reassembly of the *Ae. albopictus* genome sequence using Lep-Anchor [16] and the scaffolds of the Palatini et al. (2020) AalbF2 *Ae. albopictus* genome assembly; (3) We performed a bulk segregant analysis (BSA) of diapause incidence in the F_4_ intercross generation of two independent lines created by crossing a temperate female and tropical male and determining SNP frequencies of high- and low-diapause bulks; (4) We genotyped 9 tropical males and 11 temperate females at almost 1 million SNP loci using RNAseq; (5) We identified candidate regions of the *Ae. albopictus* genome putatively associated with diapause by identifying SNPs that differed both between tropical and temperate samples and between high- and low-diapause bulks.

### 2.2. Collection of Mosquito Strains

We used a population from Manassas, Virginia (“TEMP”, a diapausing, temperate population) and a population from Kuala Lumpur, Malaysia (“TROP”, a non-diapausing, tropical population). The TEMP colony was established with over 500 larvae collected from a local tire recycling center and maintained for seven generations under near-optimal conditions of larval nutrition, a 16L:8D photoperiod at 21 °C and approximately 80% relative humidity as described previously [24,25]. These conditions were chosen based on the previous optimization of mosquito rearing and to provide a long-day (non-diapause) control for diapause-inducing conditions (8L:16D, 21 °C). The TROP colony was established using at least 1000 eggs collected from oviposition traps and was maintained for three generations under the same conditions as the TEMP colony. Both colonies were maintained as large (census size > 300 individuals), outbred populations before initiating the intercross mating described below. The TEMP population corresponds to “VA” and “MAN” and the TROP population corresponds to “KLP” of previous publications from this laboratory [5,18,23,25].

### 2.3. Intercrossing Tropical and Temperate Lines for Linkage Mapping and BSA

The intercross lines used for the linkage mapping and BSA were established by mating an individual TROP male and an individual TEMP female for each line (Figure 1). We performed all of the crosses between tropical males and temperate females to maximize the chances of obtaining individual females with a high diapause incidence in the F_4_ intercross generation for the BSA experiment. The rationale for this decision was that some genes affecting diapause might be sex-linked. To perform the cross, sixty mating cages were established with one TROP male and three TEMP females per cage. After four days to allow for mating, TROP males were snap-frozen in liquid nitrogen and stored at −80 °C and TEMP females were individually placed into fly vials with a strip of unbleached paper towel as a substrate for oviposition. Two females that oviposited > 20 eggs each were snap-frozen in liquid nitrogen and stored at −80 °C for subsequent RNAseq genotyping. The eggs (intercross F_1_) of these females were maintained and hatched out under the near-optimal conditions described above to establish two independent intercross lines. Both lines were used for the BSA in the intercross F_4_ generation. We choose to breed to the intercross F_4_ generation for the BSA experiment because we hypothesized that additional intercross generations would lead to a decrease in the diapause response of intercross females. Additionally, one line was arbitrarily chosen and maintained to the intercross F_7_ generation to increase recombination among marker SNPs for linkage mapping (Figure 1). We choose to interbreed to the F_7_ generation for the linkage map experiment to obtaining sufficient recombination to resolve the relative positions of our high-density SNP makers without excessive multiple recombination events between markers. Additional details can be found in the Appendix A.

### 2.4. Linkage Mapping: Tissue Preparation, RNA Extraction, and Sequencing

In the intercross F_7_ generation, larvae were reared to adults, collected one week after eclosion, snap-frozen in liquid nitrogen, and individually stored at −80 °C. Total RNA was extracted from each of the two parents and 70 individual F_7_ intercross mosquitoes using a modified TRI^®^ Reagent (Sigma Aldrich, St. Louis, MO) RNA extraction protocol described in previous publications [17,19]. Samples of total RNA from two intercross F_0_ parents were used for library preparation and sequencing alongside those of other TROP and TEMP individuals as described below (see “*Bulk segregant analysis: RNA extraction and sequencing*”). Samples of total RNA from the 70 individual F_7_ offspring were used to create individual sequencing libraries using a modified protocol from the NEBNext^®^ Ultra™ RNA Library Prep Kit and NEBNext Poly(A) mRNA Magnetic Isolation Module (New England Biolabs, Ipswich, Massachusetts). Paired-end libraries were sequenced on an Illumina HiSeq 4000 instrument (read length = 150 bp) at the Institute for Genome Sciences (IGS), University of Maryland. Additional details are described in the Appendix A. Raw reads for the linkage mapping component of this study are accessible through NCBI BioProject accession number PRJNA694122 (~1.9 billion read pairs, Appendix A).

### 2.5. Linkage Mapping: Read Cleaning and Alignment to the Palatini et al. (2020) Assembly

We removed contaminant and low-quality reads and performed read trimming, using a combination of custom scripts, Trimmomatic version 0.39, and SolexaQA++ version 3.1.7.1. Scripts for all analyses are available in the Dryad repository located here: https://doi.org/10.5061/dryad.mgqnk98z4. Alignment was performed using STAR version 2.7.1a. Additional details are provided in the Appendix A.

### 2.6. Linkage Mapping: Determining Linkage Groups

Linkage mapping was conducted with Lep-MAP3 [26] as an “F_2_” population; i.e., the two F_0_ individuals that founded the intercross line were used as grandparents and the 70 (F_7_) intercross individuals were considered to be F_2_ offspring for the purposes of Lep-MAP3. As noted above, we used F_7_ intercross offspring rather than F_2_ offspring to produce a finer-scale linkage map by allowing more recombination events between markers. As F_7_ offspring will have approximately 6 times more recombination events than F_2_ offspring, the genetic distances estimated by the Lep-MAP3 pipeline will be about sixfold overestimates, for which we have corrected below. We verified that the Lep-MAP3 pipeline is robust to the use of non-F_2_ offspring in place of F_2_ offspring by simulating comparable F_7_ data and running Lep-MAP3 on these data.

SNP genotypes were determined in a likelihood framework within Lep-MAP3. First, the likelihood of each genotype for each individual was calculated with the Lep-MAP3 pipeline (pileupParser.awk + pileup2posterior.awk) using samtools mpileup [27] from individual .bam alignments (mapped to the scaffold level assembly of Palatini et al. (2020)). Then we made an F2 pedigree file with two “dummy” parents and ran the ParentCall2 module on this pedigree and the genotype likelihoods. After this, we ran the Filtering2 module with parameter dataTolerance = 0.0001 to remove markers showing distorted segregation (1:10,000 odds of the observed distortion due to chance, assuming an F_2_ population).

Using a custom script, markers were separated into 4 types based on the genotypes of the intercross F_0_ parents (1: AA female + BB male, 2: AB + AA, 3: AA + AB, 4: AB + AB), and a Lep-MAP3 map file was created for each group of markers. The module SeparateChromosomes2 was called on each of these groups (parameter map), a LOD score limit (lodLimit) between 8 and 11 and (lod3Mode = 2) was used for each group to split these markers into three major linkage groups (pseudochromosomes). Different LOD scores were used for each group because they contained a different number of SNP markers. Nearby markers were inspected to match the three linkage groups in all grandparent combinations, yielding a map with ~ 100,000 markers in 12 linkage groups, with groups 3, 6, 9, and 12 matching to the first pseudochromosome, 1, 4, 7, and 10 matching to the second pseudochromosome, and 2, 5, 8, and 11 to the third. Then the module JoinSingles2All was called with lodLimit = 10, increasing the number of markers in linkage groups to almost 183,000, with the new markers consisting mostly of markers with uncertain grandparental genotypes.

OrderMarkers2 (OM2) was then run on the 12 linkage groups. Because of the difficulty of phasing the data when both grandparents were heterozygous, we did not include these linkage groups in further analyses (linkage groups 10, 11, 12). We used the parameter grandParentalPhase = 1 in OM2 to phase data on the (homozygote) grandparents. Module OM2 was run 5 times for each of the remaining 9 groups and the solution with the highest likelihood was kept for each linkage group. Also, the marker intervals were calculated (calculateIntervals = file), indicating uncertainty of marker positions, which were used later in the analysis.

### 2.7. Genome Sequence Reassembly Using Lep-Anchor

Genome anchoring, haplotype reduction, and scaffold reassembly of the Palatini et al. (2020) AalbF2 genome assembly were achieved with Lep-Anchor [16]. The input initially consisted of the 9 linkage maps described above with marker intervals from Lep-MAP3 consisting of 111,328 SNP markers. Additionally, we used a contig-contig alignment chain file calculated using HaploMerger2 [28] and mappings from minimap2 [29] for *Ae. albopictus* PacBio reads (SRR8839546-57, 60–70). We first split the scaffold level assembly of Palatini et al. (2020) into contigs using a custom script. To utilize the scaffolding links (Hi-C) from the Palatini et al. (2020) genome, we constructed an artificial paf file with alignments between each contig joined initially into scaffolds. We followed the pipeline given in Lep-Anchor’s wiki [30] (scripts provided in our dryad repository), first removing contigs being full haplotypes with findFullHaplotypes.awk (parameter minScore = 20) based on the alignment chain only. We then mapped the markers in these haplotype contigs into non-haplotypes using the LiftoverHaplotypes module. Next, CleanMap was run to put contigs into chromosomes. CleanMap split 11 contigs into two or more chromosomes. We determined the split sites in these 11 contigs by inspecting the contig-contig alignments manually and by HaploMerger2 (hm.batchA3.misjoin_processing). A total of 5 contigs could be split exactly, and the location of the split-site could be approximated based on the alignments for 5 of the remaining 6 contigs.

We then ran PlaceAndOrientContigs (POC) with a bed file from Map2Bed including all non-haplotype contigs not assigned to any chromosome (i.e., contigs with no linkage map markers). Following POC, we used the propagate script (iterated 9 times, until additional contigs could not be added) to place contigs that could be assigned to a single chromosome. The remaining contigs were left without chromosome assignment. We ran POC three times, first with only the contigs with a clear assignment and two more times removing newly found haplotypes between runs (using removeHaplotypes.awk).

Custom R scripts were used to obtain Marey maps for each chromosome. This pipeline was run twice: after the first run, we identified regions in three of the maps that contradicted the other maps (one in pseudochromosome 2, and two in pseudochromosome 3). The contradictory regions were all in those maps produced using markers for which one intercross F_0_ parent was a heterozygote. The problematic regions produced by the first pipeline run seemed to be caused by gaps without markers (and a high amount of crossing over due to the F_7_ intercross design). By flipping markers in one region (of linkage map 5) and splitting two other maps into 2 and 6 parts (linkage maps 4 and 8, respectively) we could obtain a set of 15 consistent maps that were used in the second run of the pipeline.

### 2.8. Genome Reassembly: Analysis of Lep-Anchor Results

To compare the genome sequence reassembly produced by Lep-Anchor (AalbF3) to the AalbF2 assembly of Palatini et al. (2020), we calculated genome size and GC content, the number and size of scaffolds and contigs, maximum scaffold length, and the percentage of each genome in scaffolds of 50 kb or more. We also compared the two genome assemblies using BUSCO version 4.0.4 [31] using the “Diptera odb10” dataset of single-copy genes and default configurations.

To compare the annotations of the two genomes, we used custom scripts to analyze the annotation of AalbF2 accessed through the NCBI *Ae. albopictus* Annotation Release 102 [32]. Using custom scripts, we built an annotation for the AalbF3 reassembly out of the NCBI annotation as follows: for each feature annotation, we determined whether or not the entirety of the annotation fell in a contig of AalbF3. If it did, we calculated its new position on the scaffolds of AalbF3, and included the re-mapped feature in our new annotation. We also compared the number of feature annotations in the two assemblies to the number of features in the NCBI *Ae. aegypti* Annotation Release 101 [33].

Finally, we aligned the transcriptome sequences from 11 male TROP individuals and 9 female TEMP individuals to both the AalbF2 assembly and the AalbF3 reassembly. The transcriptome sequences were cleaned and filtered as described in Appendix A. Both genomes were indexed using the bowtie2-build tool from bowtie2 version 2.3.5.1 [34]. We then aligned the cleaned and filtered reads to each genome using the default (end-to-end) settings of the bowtie2 aligner. We used the bowtie2 aligner with end-to-end settings for this analysis because it is a commonly used algorithm for short-read alignment that produces easily interpretable results. Although overall alignment rates might be higher using a split-read aligner such as HISAT2 or STAR, because only 11 contigs from the AalbF2 assembly were cut to produce the AalbF3 reassembly (see above), using end-to-end alignments should not affect the comparison of alignment rates between the two assemblies.

### 2.9. Lep-Anchor Genome Sequence Reassembly: Establishing Chromosome Identity

A preliminary analysis indicated that approximately half of pseudochromosome 3 of AalbF3 was inverted relative to its true orientation. We, therefore, inverted the scaffolds in this region to align them with the physical mapping of AalbF2 and with the syntenic region of the *Ae. aegypti* chromosome 3. We note that our linkage map contained supporting markers for both orientations of this inverted part of pseudochromosome 3.

To further determine the correspondence between the pseudochromosomes of the AalbF3 reassembly and the *Ae. albopictus* physical chromosomes, we took two approaches. First, we used the sequence of 50 fluorescence in situ hybridization (FISH) probes used by Palatini et al. (2020) to link scaffold positions in AalbF2 to relative positions on the physical chromosomes. For the 33 of their 50 probes which mapped to contigs that were included in AalbF3, we used custom scripts to determine the corresponding position on each of our pseudochromosomes. Of the remaining 17 probe sequences, we were able to assign 16 unambiguously to a position on both AalbF3 and the *Ae. aegypti* assembly (AaegL5.0) using BLASTn (version 2.9.0+) searches that identified a single high bit-score match (bit-score >2000).

Second, we analyzed synteny between each pseudochromosome and the chromosomes of *Ae. aegypti.* We downloaded the AaegL5.0 protein set (GCF_002204515.2) and the *Ae. albopictus* AalbF2 protein set (GCF_006496715.1) and identified putative orthologues as reciprocal best hits [35] using BLASTp (version 2.9.0+) with the following thresholds: e-value = 1e-20, percent identity = 85%, percent query sequence coverage = 50%. We then used custom scripts to filter the AaegL5.0 annotation file [33] and the annotation file for AalbF3 (see above) to determine the position of each putatively orthologous gene on each species’ respective chromosomes.

We also discovered that the male-determining gene *Nix* was found on a scaffold belonging to pseudochromosome 3, whereas *in-situ* hybridization has shown it to be located on chromosome 1 [36]. Further investigation of this scaffold suggested that it was chimeric: the 5’ contigs of the scaffold had a single putative ortholog gene from our previous analysis, which mapped to chromosome 3 of *Ae. aegypti*; the 3’ contigs of the scaffold had two putative ortholog genes, both of which mapped to chromosome 1 of *Ae. aegypti*, and this region also contained *Nix.* The results of the synteny analysis with *Ae. aegypti* described above showed that chimeric contigs such as this one are extremely rare in our analysis. We, therefore, moved *Nix* to its inferred position on pseudochromosome 1 by cutting the scaffold (scaffold 3.129), leaving 2,225,968 bases on the 5’ end (which matched to chromosome 3 of *Ae. aegypti*) in place, and moving the remaining bases of the 3’ end to pseudochromosome 1, assigning it a position based on the synteny of the two orthologous genes on this (sub-) scaffold with *Ae. aegypti*. We also used custom scripts to modify the accompanying annotation file to account for the changes in the corresponding assembly.

### 2.10. Bulk Segregant Analysis (BSA): Measuring Diapause Phenotypes

The diapause phenotypes of individual females from the two F_4_ intercross lines were determined to create bulks of individuals with extreme phenotypes for subsequent SNP genotyping using RNAseq. Here, we define the diapause phenotype of an individual female as diapause incidence (DI), the proportion of diapause eggs produced by a female maintained under an unambiguous short-day photoperiod (8L:16D) at 21 °C, conditions that produce the optimal expression of diapause in *Ae. albopictus* [25,37]. F_4_ intercross eggs from the two BSA lines were hatched and larvae were reared under near-optimal conditions as described above. Experimental details are provided in the Appendix A. For the first line, diapause incidence was measured for 60 females with 20 to 64 eggs per female. For the second line, diapause incidence was measured for 78 females with 22 to 69 eggs per female. For both lines, females with eggs that had a DI ≤ 10% (line 1) or 5% (line 2) were assigned to the low-diapause bulk, and females with eggs that had a DI ≥ 50% were assigned to the high-diapause bulk (Appendix A). Thus, a total of four bulks of adult females were produced based on the diapause incidence of their eggs: a low-diapause bulk (*n* = 11 females) and a high diapause bulk (*n* = 11) from the first line, and a low-diapause bulk (*n* = 12 females) and a high-diapause bulk (*n* = 8 females) from the second.

### 2.11. Bulk Segregant Analysis: RNA Extraction and Sequencing of Bulks, TEMP, and TROP Individuals

Total RNA was extracted from the 9 individual TROP males and 11 individual TEMP females, including the parents, used to produce the intercross lines, as described above (see "*Linkage mapping: RNA extraction and sequencing*”). For the four bulks, total RNA was extracted from individual F_4_ females and assessed on an Agilent chip as described above. Due to relatively low RNA extraction yields from the individual females comprising the bulks, it was necessary to pool equal amounts of RNA from females within each bulk. This produced four bulk RNA samples (low- and high-diapause from each line, see Appendix A). Illumina paired-end mRNA-Seq library construction was performed on the TEMP and TROP individual and bulk RNA samples according to the TruSeq RNA sample preparation kit (Version 2) (Illumina Inc., San Diego, CA, USA). The 20 individual TEMP and TROP libraries were individually barcoded and the 4 bulk libraries were individually barcoded according to the manufacturer’s instructions. Illumina paired-end sequencing was conducted by IGS at the University of Maryland. Additional details are provided in Appendix A. Raw reads for the BSA, TEMP and TROP sequencing components of this study are accessible through NCBI BioProject accession number PRJNA694122 (~1.2 billion read pairs, Appendix A).

### 2.12. Bulk Segregant Analysis: Cleaning and Aligning Transcriptomes

All sequencing libraries from the TEMP, TROP, and bulk samples were cleaned and aligned to the AalbF3 genome assembly using the steps described for linkage mapping (See “*Linkage mapping: cleaning and aligning transcriptomes*”).

### 2.13. Bulk Segregant Analysis: SNP Calling and Filtering

We identified single-nucleotide polymorphisms (SNPs) in a joint-genotyping context [38] using the GATK pipeline, version 4.1.4.0, largely following GATK best practices [39]. Details are provided in Appendix A. After SNP calling and initial quality filtering, we further filtered our SNP set to include only SNPs which were genotyped in all four bulks (i.e., both the high-diapause and low-diapause bulks from both lines) and at least five individuals from the temperate population and at least five individuals from the tropical population. For each SNP, we calculated the difference in reference allele frequency (referred to hereafter as Allele Frequency Difference, or AFD [40] for three comparisons: (1) between the (diapausing) TEMP and (non-diapausing) TROP individuals, (2) between the high-diapause and low-diapause bulks of the first BSA line, and (3) between the high-diapause and low-diapause bulks of the second BSA line.

### 2.14. Bulk Segregant Analysis: Identifying Putative Diapause-Associated SNPs

We first filtered potentially diapause-associated SNPs by identifying those for which all three allele frequency differences (AFDs) had the same sign (i.e., TEMP vs. TROP, high-diapause vs. low-diapause bulks for both BSA lines). From among the resulting 46,736 SNPs, we then calculated a diapause associated *p*-value and false discovery threshold as described in detail in Appendix A. Briefly, for each SNP, a *p*-value was calculated as the percentile |AFD| between the TEMP and TROP samples. As the maximum possible |AFD| depends on the minor allele frequency (MAF), we binned the SNPs by MAF as described in the Appendix A, and then calculated percentiles within each bin. Then, again for each SNP, two additional *p*-values were calculated as the percentile |AFD| between the high-diapause and low-diapause bulks for each BSA line, contingent upon the parental genotype at that site. The contingency was applied since SNPs for which the parents had two of each allele are more likely to become strongly differentiated solely by chance than SNPs for which the parents have one reference allele and three alternate alleles. Finally, all three *p*-values were multiplied to produce an overall diapause-associated *p*-value. We then established three *p*-value false-discovery thresholds to account for the testing of multiple SNPs. These thresholds correspond to an expectation of 0.05, 1, and 5 false-positive SNPs based on the set of 46,736 SNPs described above. Finally, we identified putative diapause-associated SNPs as those that meet the following two criteria. First, an |AFD| > 0.5 between the TEMP and TROP samples, indicating that the SNP is segregating in nature. Second, a *p-*value less than the *p*-value false-discovery threshold corresponding to 0.05, 1, and 5 false-positive SNPs (see Appendix A).

Finally, we used custom scripts and the annotation file to identify genes (including long non-coding RNAs, lncRNA) and pseudogenes within 50 kb of the diapause-related SNPs at each false-discovery threshold.

## 3. Results

### 3.1. Linkage Mapping: Transcriptome Sequencing

Individual RNAseq of the F_0_ TEMP female and TROP male parents and 70 intercross F_7_ offspring produced between 20–60 million paired-end reads per individual, leading to 111,328 high-quality SNPs that were used for creating a high-density linkage map. (Appendix A).

### 3.2. Linkage Mapping: Determining linkage Groups

We initially identified 182,503 SNP markers on 3 pseudochromosome linkage groups. Of these, 111,328 could be phased based on the intercross F_0_ TEMP female and TROP male parents that were single-pair mated to produce the mapping line. These markers were ordered within linkage groups and used to reassemble the contigs of the Palatini et al. (2020) genome assembly (AalbF2) using Lep-Anchor [16]. The final resulting linkage map included 573 scaffolds comprising 3 pseudochromosomes of length 587 Mb, 491 Mb, and 372 Mb (total size = 1.45 Gb). Since we used the F_7_ generation for mapping instead of the F_2_ generation, we accounted for the increased number of recombination events by dividing the genetic distances by 6 to produce genetic lengths of approximately 50 cM for the three chromosomes. Due to the possibility of undetected multiple-crossover events between markers, these estimates should be considered a lower boundary of the true value. Marey plots for each chromosome showing the relationship between physical distance and recombination distance for each pseudochromosome are given in Figure 2 and Appendix A. Flat, low-recombination regions in the center of each chromosome indicate the centromeres, which are metacentromeric or submetacentromeric in *Ae. albopictus* [41].

### 3.3. Genome Sequence Reassembly Using Lep-Anchor: Analysis of Results

The AalbF3 assembly has been deposited at DDBJ/ENA/GenBank under the accession JAFDOQ000000000. The version described in this paper is version JAFDOQ010000000. Basics statistics comparing the size and contiguity of the AalbF2 and AalbF3 assemblies are given in Table 1. The AalbF3 reassembly was 57% of the length of AalbF2. Of the 1.09 Gb removed from AalbF2, 0.2 Gb were removed as a part of contigs that could not be connected to the linkage maps and this sequence may contain biologically important non-transcribed elements of the genome; the remaining 0.89 Gb were identified as haplotypes of sequences in the final reassembled genome and were collapsed accordingly. While the AalbF3 reassembly was 57% of the length of AalbF2, AalbF3 contained only 26% fewer scaffolds and 20% fewer contigs. The contigs that were incorporated into the AalbF3 reassembly were longer than the AalbF2 contigs (7.5 Mb contig N50 compared to 1.2 Mb). In contrast, the scaffolds of the AalbF3 reassembly were smaller than those in the AalbF2 assembly (10 Mb scaffold N50 compared to 56 Mb scaffold N50). This is because we did not join contigs or scaffolds into longer scaffolds if the relative orientation or position was unknown within the linkage map. Concatenating our scaffolds would yield almost full chromosomes but would induce many short-scale structural errors. Moreover, these short scaffolds enable further scaffolding and improvements with new sequencing and/or linkage data. The proportion of the genome found in very small (less than 50 kb) scaffolds declined somewhat from 0.47% in AalbF2 to 0.03% in AalbF3.

The results of BUSCO analyses of the two genome assemblies are presented in Table 2. The reduced genome size of AalbF3 reduces the number of putatively-artifactual duplications almost by half, from 16.6% complete and duplicated BUSCO genes to only 8.9%. In contrast, the number of complete BUSCO genes declined only slightly, from 94.6% in AalbF2 to 93.2% in AalbF3. Furthermore, the proportion of complete and single-copy genes increased from 78% in AalbF2 to 84.3% in AalbF3. About half of missing complete BUSCO genes in AalbF3 occur in a fragmented form; the proportion of fragmented BUSCO genes rose slightly, from 4.2% in AalbF2 to 4.9% in AalbF3. The small increase in missing and fragmented BUSCO genes in AalbF3 is likely because some of the contigs removed during re-assembly contained genes and/or exons.

A comparison of the annotated features in the AalbF2 and AalbF3 assemblies is presented in Table 3. As noted above, a total of 57% of the bases of AalbF2 were removed in AalbF3 reassembly. However, many of the removed sequences had few annotated features, as demonstrated by the fact that, with the exception of pseudogenes, 72–83% of the annotated features were retained in AalbF3. The annotated features from AalbF2 that were not included in AalbF3 were likely artifactually duplicated in the AalbF2 assembly, an interpretation supported by the decrease of complete and duplicated BUSCO genes in AalbF3 relative to AalbF2 (Table 2). All classes of annotated features have more annotations in AalbF3 than in the *Ae. aegypti* assembly AaegL5.0.

Finally, we tested how well *Ae. albopictus* transcriptome sequences from the 20 TEMP and TROP intercross F_0_ samples aligned to both genome assemblies. The overall alignment rate of the paired-end reads to AalbF2 was 81% (±2% standard deviation). Of these aligned reads, 22% (±1%) of reads aligned concordantly to a single location in the genome, while 44% (±2%) of reads aligned concordantly to multiple locations. The overall alignment rate of reads mapped to AalbF3 was 79% (±2%). Of these aligned reads, 46% (±4%) of reads aligned concordantly to a single location in the assembly, and 19% (±4%) aligned concordantly to multiple locations. Thus, although the overall alignment rate was reduced slightly by using AalbF3 as a reference for read mapping, the proportion of aligned reads with unambiguous mapping more than doubled.

### 3.4. Genome Reassembly: Establishing Chromosome Identity

Mapping the physical markers (FISH probe sequences) of Palatini et al. (2020) to the AalbF3 reassembly allowed us clearly to establish the identity of each pseudochromosome. Between 8–21 FISH probe sequences from a single true chromosome mapped to each pseudochromosome (Figure 3). We, therefore, refer to the pseudochromosomes of our *Ae. albopictus* AalbF3 assembly as Chromosomes 1, 2, and 3 hereafter. These markers map in the expected arrangement and order on each chromosome, with some minor exceptions. In some cases, the order of two markers is reversed both in AalbF3 and in the *Ae. aegypti* assembly relative to the AalbF2. For example, in both assemblies, the probe assigned to 1q33 is proximal compared to the markers assigned to 1q31. While it is possible that AalbF3 and the *Ae. aegypti* assembly are both misassembled in the same way at these sites, it is also possible that in these cases the probes were misassigned to their particular chromosome bands, and that their order in our assembly is correct. The three cases where the order of the probes on AalbF2 given in Palatini et al. (2020) agrees with the order in the *Ae. aegypti* assembly, but not with AalbF3, are more likely to represent errors in our assembly. These cases of likely error include two markers to 1q25 found between two other markers to 1q22 in AalbF3, but not in the *Ae. aegypti* assembly. Also, a marker to 2q26 is found between two other markers to 2q24. Finally, two markers to the base of the q arm of chromosome 3 (3q11 and 3q13) of AalbF2 are instead found among markers at the base of the 3p arm of AalbF3.

We found 7586 putative single-copy orthologs based on BLAST reciprocal best hits which could be mapped to both the *Ae. albopictus* and *Ae. aegypti* genome assemblies. The relative position of these genes in the *Ae. aegypti* genome and the *Ae. albopictus* AalbF3 reassembly established clear broad-scale synteny between pseudochromosomes 1, 2, and 3 of AalbF3 and chromosomes 1, 2, and 3 of *Ae. aegypti* (Figure 3). We note that this analysis is less reliable for assessing local synteny due to possible rearrangements of gene order between *Ae. aegypti* vs. *Ae. albopictus* and the fact that some scaffolds have relatively few genes.

### 3.5. Bulk Segregant Analysis: RNA Extraction and Sequencing

Individual RNAseq of 9 TROP males and 11 TEMP females (including the parents of the intercross lines used for the linkage map and BSA, Figure 1) produced between 11 and 27 million paired-end reads per individual (average = 19 million: Appendix A). RNAseq of the four pooled bulks (low-diapausing and high-diapausing pools from two intercross lines, Figure 1), produced between 177–208 million paired-end reads per bulk, (average = 193 million: Appendix A).

### 3.6. Bulk Segregant Analysis: Calling SNPs

Across all libraries (bulks and parental populations alike), a total of 9.8 million SNPs were identified that passed our quality filters. Of these, 957,000 SNPs were present in all four bulks and 5 or more individuals from both the tropical and temperate populations.

### 3.7. Bulk Segregant Analysis: Identifying Putative Diapause-Associated SNPs

Of the 957,000 SNPs found in all three comparisons (TROP vs. TEMP and high-diapause bulk vs. low-diapause bulk in two BSA lines), 46,736 SNPs had a non-zero allele frequency difference of the same sign in all three comparisons. Allowing for false discovery thresholds of 0.05, 1, and 5 SNPs, and including only those SNPs with |AFD| > 0.5 between the TEMP and TROP populations, produced 4, 77, and 260 putative diapause associated SNPs at the three false discovery thresholds, respectively (Table 4, Figure 4, Appendix A). These SNPs are found in many locations throughout the genome, but they are largely clustered toward the beginning of chromosome 1, the end of chromosome 2, and the beginning and end of chromosome 3 (Figure 4). These regions with many putative diapause-associated SNPs also have elevated levels of allele frequency difference across all sites (Appendix A). This elevated allele frequency difference in these chromosomal regions is found in all three comparisons but is strongly driven by the comparison of mosquitoes from TEMP and TROP populations (Appendix A). Lists of the annotated features found within 50 KB of SNPs identified under all three false-discovery thresholds are provided in Appendix A. The gene models within 50 KB of SNPs at the 0.05 FDR threshold are *gametogenetin-binding protein 2-like*, *segmentation polarity homeobox protein engrailed-like*, *casein kinase II subunit alpha*, *homeobox protein OTX2-A*, *ice-structuring glycoprotein*, *3-ketodihydrosphingosine reductase*, *methyl-CpG-binding domain protein 3*, *vacuolar protein sorting-associated protein 45*, *nuclear pore complex protein Nup58*, *metaxin-1 homolog*, and one uncharacterized protein.

## 4. Discussion

Producing a high-quality genome sequence for *Ae. albopictus* represents a significant challenge due to the large size and abundance of repetitive DNA in the genome of this mosquito [10,15]. Our approach employed RNAseq as a method for reduced-representation sequencing to identify SNP markers for linkage map construction. This method succeeded in producing 111,328 informative SNP markers spread throughout the *Ae. albopictus* genome. We also used an unconventional F_7_ intercross design to construct the linkage map, rather than the more common F_2_ design. This approach has the potential drawback that it increases the probability of multiple, and therefore undetected, recombination events between markers; for this reason, the approximately 50 cM length of each chromosome is likely an underestimate. This conclusion is consistent with the results of a previous linkage map analysis of *Ae. albopictus* based on 73 single-strand conformation polymorphism cDNA markers, which identified three linkage groups of 64.5, 76.5, and 71.6 cM [42]. Nevertheless, the F_7_ approach increases the number of recombination events and therefore allows for finer-scale mapping, which was valuable in using the high-density linkage map as a basis for the Lep-Anchor reassembly (see below). Our strategy was successful in producing three chromosome-sized linkage groups with presumably-centromeric regions of reduced recombination (Figure 2), consistent with the *Ae. albopictus* karyotype [41].

We used our high-density linkage map to perform Lep-Anchored reassembly of contigs from the AalbF2 assembly produced by Palatini et al. [15]. We refer to the resulting reassembly as AalbF3. The rationale for using Lep-Anchored reassembly is that the map positions of the high-density linkage markers can be used to remove haplotigs, as well as order contigs and scaffolds and put them into chromosomes [16]. Indeed, two lines of evidence indicate that our reassembly was largely successful in correctly ordering the scaffolds. First, the FISH probes described in Palatini et al. [15] map to the correct chromosomes of AalbF3, and in the expected order, with only a few, small scale exceptions. Second, synteny of the *Ae. albopictus* AalbF3 chromosomes with *Ae. aegypti* was extremely strong, even at scales as small as 10–20 Mb (Figure 3). Although the precise order and spacing of all 573 scaffolds to each other is not known with certainty, these results indicate that the overall ordering of these scaffolds by the linkage map is accurate.

The AalbF3 reassembly is an improvement relative to the previous AalbF2 assembly in several respects. First, cytometric estimates [43,44] and comparisons to the *Ae. aegypti* genome suggests that the size of the AalbF2 assembly (2.54 Gb) is two times larger than expected, likely due to the presence of haplotypes incorrectly determined to be unique contigs during assembly. This inference is further supported by the identification of up to 1329 likely artifactually duplicated gene copies in AalbF2 [15]. The Lep-Anchor reassembly process identified and discarded 43% of AalbF2, bringing the total genome size to 1.45 Gb (Table 1), which is much closer to the cytometric estimates of 1.19–1.28 Gb [15]. Second, while BUSCO analysis indicates a slightly lower percentage of complete BUSCO genes in AalbF3 (93.2%) relative to AalbF2 (94.6%), the number of complete and single-copy BUSCO genes is higher in AalbF3 (84.3%) relative to AalbF2 (78%). Also, the number of complete and duplicated BUSCO genes is lower in AalbF3 (8.9%) relative to AalbF2 (16.6%) (Table 2). Finally, the proportion of aligned RNAseq reads that mapped concordantly to a single position of AalbF3 was 46%, a substantial improvement relative to 22% for the AalbF2 assembly.

Despite the significant advantages of AalbF3 described above, AalbF2 may provide a more suitable reference assembly for certain genomic studies, such as those focused on certain classes of non-coding regions of the genome. By taking into account long-read data, contig-contig alignments, and previous (Hi-C) scaffolding links, Lep-Anchor can assemble many contigs that do not have any markers on them, but are linked by other data (e.g., PacBio reads). Nevertheless, approximately 200 Mb of AalbF2 was discarded during reassembly due to lack of linkage-map markers and linking PacBio reads. Because we used RNAseq-derived SNP data to construct the linkage map used for reassembly, the discarded contigs are likely very gene-poor but may contain biologically important non-transcribed sequences. We provide a list of all the discarded contigs from AalbF2 in our dryad repository.

We used the improvements to the *Ae. albopictus* reference genome assembly (AalbF3) to identify chromosomal regions putatively associated with diapause by combining two sources of evidence. First, we used a bulk segregant analysis (BSA) in the intercross F_4_ generation of two experimental lines created by crossing a single TEMP female to a single TROP male. Our approach was to identify SNPs with extreme allele frequency differences between high-diapause and low-diapause bulks in each line. Second, we used the comparison of genome-wide SNP frequencies between 11 females from the TEMP population, which has a robust photoperiodic diapause response, and 9 males from the TROP population that is genetically incapable of diapause [22,23]. Calculating a combined *p*-value and applying the strictest correction for false-discovery indicated 4 SNPs distributed across all three chromosomes that were highly differentiated in both comparisons: the TROP vs. TEMP populations that represent the frequency of these SNPs in colonies recently derived from natural populations, and the two bulk segregant analyses that show SNP differentiation associated with phenotypic extremes of the experimental intercross lines (Figure 4, Appendix A). The concordant elevated |AFD| in both the BSA and TEMP vs. TROP comparison at the beginning of chromosome 1, the end of chromosome 2, and both the beginning and end of chromosome 3 are particularly notable. Decreasing the stringency of the false-discovery rate resulted in the identification of 77 and 260 SNPs, many more than expected due to false discovery caused by multiple tests (1 and 5, respectively). The SNPs identified at FDR thresholds of 1 and 5 tended to have a smaller level of differentiation between the different comparisons (Appendix A), suggesting that they may have a smaller effect size on the diapause phenotype than those SNPs identified under more stringent FDR threshold of 0.05.

Despite several notable clusters of concordant elevated |AFD| noted above, the overall distribution of candidate SNPs on all three chromosomes implies a polygenic architecture underlying the evolution of photoperiodic diapause in *Ae. albopictus* (Figure 4). This is consistent with a number of recent studies that have used high-throughput sequencing approaches, including RNAseq and genome-wide SNP analyses, to probe the genetic basis of diapause in other species [45,46,47]. The list of gene models located within 50 Kb of SNPs at three FDR thresholds (Appendix A) includes a number of intriguing candidates, including genes implicated in cell-cycle regulation (*gametogenetin-binding protein 2-like*), epigenetic regulation (*methyl-CpG-binding domain protein 3*), and cold tolerance (*ice-structuring glycoprotein*). The genes in Appendix A do not overlap substantially with genes identified in previous studies comparing transcriptional changes associated with diapause in *Ae. albopictus* [5,17,18]. We hypothesize that this is because the earlier transcriptional studies focused on tissues and developmental stages related to the proximal physiological basis of diapause (i.e., “downstream” processes). The candidate SNPs identified in this study are presumably related to the evolution of this complex adaptation from a tropical, ancestral population incapable of undergoing photoperiodic diapause, to a temperate, derived population that relies on diapause to survive temperate winters [21,22,48]. Thus, the chromosomal regions identified in this analysis will provide a basis for more targeted studies investigating the functional and evolutionary genetics of genes that may be involved in the more “upstream” regulation of diapause in *Ae. albopictus*. In the longer term, characterizing the genetic basis of diapause may provide a foundation for novel vector control strategies based on the disruption of this crucial ecological adaptation.

## Figures and Tables

**Figure 1 insects-12-00167-f001:**
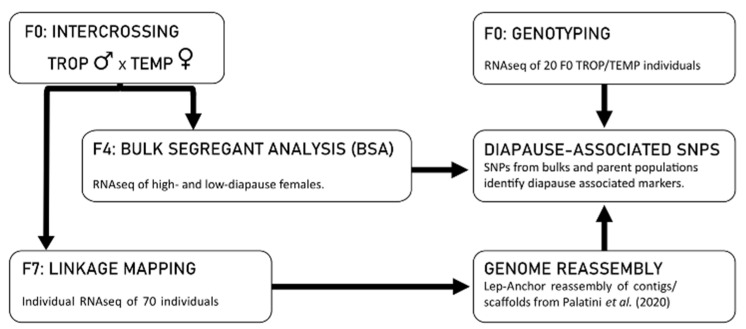
Graphical summary of this study. Two independent intercross lines were produced by mating a single TROP male to a single TEMP female. Both lines were used in the F4 BSA, with the high- and low- diapause females pooled separately for each line to produce bulks. One of the two intercross lines was propagated to the F7 for linkage mapping. RNAseq was performed to identify SNPs in the F4 BSA lines, the F7 linkage mapping individuals, as well as nine TROP males and eleven TEMP females from the F0 generation. SNPs from the TEMP individuals, the TROP individuals, and the bulks were used to identify putative diapause-associated SNPs. F0, F4, and F7 in the figure refer to intercross generations.

**Figure 2 insects-12-00167-f002:**
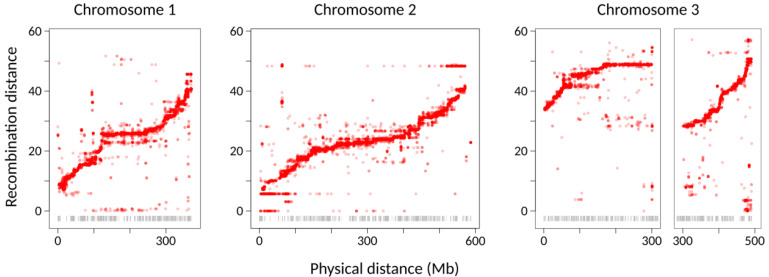
Linkage maps for the three *Ae. albopictus* pseudochromosomes. Each point shows the physical (*x*-axis) and recombination (*y*-axis) distance of a single SNP marker from one end of the linkage group. The gray ticks at the bottom show the boundaries between contigs. The first facet in the Chromosome 3 panel indicates the region at which the anterior chromosome arm was cut and inverted. These plots show those markers for which both F_0_ parents were homozygous for different alleles. The linkage maps based on other marker types are shown in Appendix A.

**Figure 3 insects-12-00167-f003:**
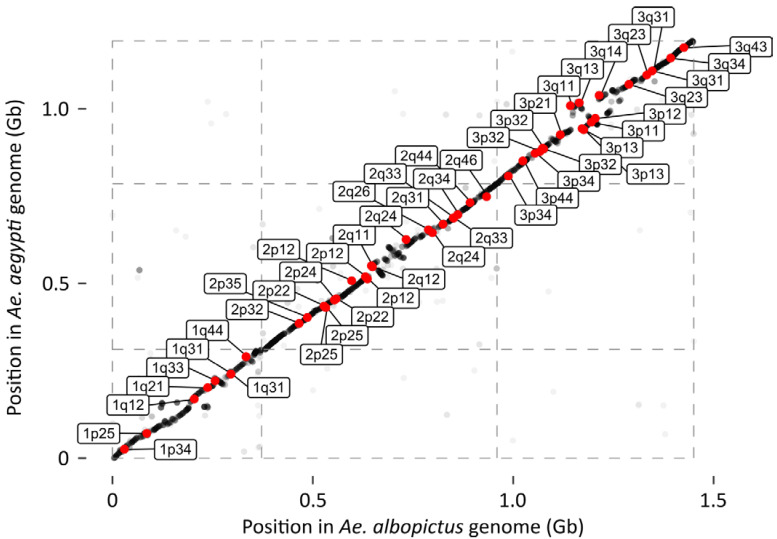
Strong synteny between the pseudochromosomes of the *Ae. albopictus* Lep-Achored genome assembly (AalbF3) and the chromosomes of *Ae. aegypti*. Each point represents the position of a putatively single-copy, orthologous gene annotation in the *Ae. albopictus* (*x*-axis) and *Ae. aegypti* (*y*-axis) genomes. Red points show where FISH probe sequences used in Palatini et al. (2020) fall in the two genomes; the label gives the positions of each probe relative to banding patterns. These probe sequences identify each of the three chromosomes unambiguously, and largely fall in the expected order.

**Figure 4 insects-12-00167-f004:**
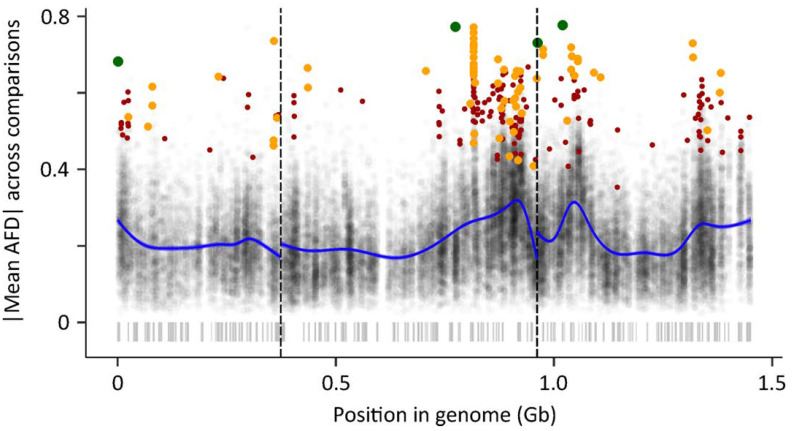
Putative diapause associated SNPs are found in clusters throughout the *Ae. albopictus* genome. Each gray point represents one of the 46,736 SNPs which had an allele frequency difference (AFD) among the high-diapause vs. low-diapause population/bulks in the same direction in all three comparisons (see text for details). The position of each point in the genome is shown on the *x*-axis (vertical dashed black lines indicate the boundaries between chromosomes). The *y*-axis shows the average AFD across all three comparisons, which was transformed to an absolute value. Blue lines show the smoothed trend of |AFD| across the genome. Colored points indicate those SNPs that are differentiated (|AFD| > 0.5) between the parent populations and pass a false-discovery threshold of 0.05 (large green points), 1 (orange points), or 5 (small red points). Scaffold boundaries are shown by the gray ticks beneath the points. While these SNPs are found in many locations across all three chromosomes, they are clustered at the beginning of chromosome 1, the end of chromosome 2, and both the beginning and the end of chromosome 3. The average allele frequency difference across all SNPs is also strongly elevated in these regions.

**Table 1 insects-12-00167-t001:** Comparison of the Palatini et al. (2020) AalbF2 assembly and Lep-Anchored reassembly AalbF3.

	Palatini et al. (2020) Assembly, AalbF2	Lep-Anchored Reassembly, AalbF3
Total size	2.54 Gb	1.45 Gb
GC%	40.40%	42.06%
Number of scaffolds	2196	573
Scaffold N50	55.7 Mb	10.1 Mb
Scaffold N75	4.2 Mb	3.5 Mb
Scaffold L50	13	43
Scaffold L75	59	99
Longest scaffold	196 Mb	43.8 Mb
Genome in 50 kb+ scaffolds	99.53%	99.97%
Number of contigs	5555	1134
Contig N50	1.2 Mb	7.5 Mb
Contig N75	0.5 Mb	2.3 Mb
Contig L50	434	56
Contig L75	1314	142

GC%: the percentage of known bases that are guanine or cytosine. N50 (N75): 50% (75%) of the genome is contained in scaffolds or contigs of this length or longer. L50 (L75): 50% (75%) of the genome is contained in this many of the longest reads.

**Table 2 insects-12-00167-t002:** BUSCO analysis of AalbF2 and AalbF3.

BUSCO Category	Palatini et al. (2020) Assembly, AalbF2	Lep-Anchored Reassembly, AalbF3
Complete BUSCO genes	3108 (94.6%)	3062 (93.2%)
Complete and single-copy	2562 (78.0%)	2769 (84.3%)
Complete and duplicated	546 (16.6%)	293 (8.9%)
Fragmented BUSCO genes	40 (1.2%)	62 (1.9%)
Missing BUSCO genes	137 (4.2%)	161 (4.9%)
Total BUSCO genes searched:	3285	3285

**Table 3 insects-12-00167-t003:** The Lep-Anchored reassembly (AalbF3) retains the most annotation features.

Feature Type	Genome			Proportion Kept in the New Genome
	AalbF2	AalbF3	*Aedes aegypti*	
mRNA	40,073	30,626	28,304	0.76
gene ^1^	36,386	26,734	19,203	0.73
lncRNA	7609	5507	4044	0.72
pseudogene ^2^	4108	2441	382	0.59
tRNA	1936	1353	910	0.70
rRNA	437	327	n/a ^3^	0.75
snRNA	143	116	n/a ^3^	0.81
snoRNA	29	24	n/a ^3^	0.83

^1^ Including both “protein-coding” and “non-coding” categories of gene annotations. ^2^ For *Ae. albopictus*, including both “transcribed pseudogenes” and “non-transcribed pseudogenes” categories of annotations. The report for *Ae. aegypti* lists only the category “pseudogenes”. ^3^ These annotation categories are not included in the report for *Ae. aegypti*.

**Table 4 insects-12-00167-t004:** Putative diapause-associated SNPs identified at three false-positive thresholds.

False Positive Threshold	0.05 SNPs	1 SNP	5 SNPs
Number of diapause-related SNPs	4	77	260
Scaffolds with SNPs	5	33	67
Nearby ^1^ protein coding genes	14	149	451
Nearby lncRNA	3	19	47
Nearby pseudogenes	2	15	38
Other nearby annotations ^2^	0	3	13

^1^ Annotations within 50 kb of a SNP were considered “nearby”. ^2^ Including tRNA and miscellaneous RNA annotations.

## Data Availability

Raw reads for the linkage mapping and BSA components of this study are accessible through NCBI BioProject accession number PRJNA694122. The AalbF3 assembly has been deposited at DDBJ/ENA/GenBank under the accession JAFDOQ000000000. The version described in this paper is version JAFDOQ010000000. Scripts for all analyses are available in the Dryad repository located here: https://doi.org/10.5061/dryad.mgqnk98z4.

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
