# Peer review of "A Linkage-Based Genome Assembly for the Mosquito *Aedes albopictus* and Identification of Chromosomal Regions Affecting Diapause"

_insects, 2021, doi:10.3390/insects12020167_

Round 1
Reviewer 1 Report
You have identified chromosome regions that you state are associated with diapause. Although this seems probable, considering the lines used, it is still an association so wouldn't it be safer to qualify this statement by saying "possible association" or something similar?
Could you indicate which specific protein coding genes are near the FPT 0.05 SNPs? There are only 11 on 4 scaffolds.
You state in the Discussion (614-617) That the gene models include a wide-range of 'intriguing candidates'. Could you indicate in the manuscript which ones you think are the most relevant?
There are a few minor 'typos' but these will probably be picked up in a future edit.
Author Response
REVIEWER #1:
Comment #1: You have identified chromosome regions that you state are associated with diapause. Although this seems probable, considering the lines used, it is still an association so wouldn't it be safer to qualify this statement by saying "possible association" or something similar?
Response #1: We have changed “associated with photoperiodic diapause” to “putatively associated with photoperiodic diapause” in the abstract and throughout the text of the manuscript.
Comment #2: Could you indicate which specific protein coding genes are near the FPT 0.05 SNPs? There are only 11 on 4 scaffolds.
Response #2: We have added the following text to lines 559-564 (results section):
“The gene models within 50KB of SNPs at the 0.05 FDR threshold are gametogenetin-binding protein 2-like, segmentation polarity homeobox protein engrailed-like, casein kinase II subunit alpha, homeobox protein OTX2-A, ice-structuring glycoprotein, 3-ketodihydrosphingosine reductase, methyl-CpG-binding domain protein 3, vacuolar protein sorting-associated protein 45, nuclear pore complex protein Nup58, metaxin-1 homolog, and one uncharacterized protein.”
Comment #3: You state in the Discussion (614-617) That the gene models include a wide-range of 'intriguing candidates'. Could you indicate in the manuscript which ones you think are the most relevant?
Response #3: We have added the following text to lines 668-672:
“The list of gene models located within 50Kb of SNPs at three FDR thresholds (SI, Table S4) includes a number of intriguing candidates, including genes implicated in cell-cycle regulation (gametogenetin-binding protein 2-like), epigenetic regulation (methyl-CpG-binding domain protein 3), and cold tolerance (ice-structuring glycoprotein).”
Comment #4: There are a few minor 'typos' but these will probably be picked up in a future edit.
Response #4: We have identified and corrected a small number of typos in the course of our revisions.
Reviewer 2 Report
The Asian tiger mosquito Aedes albopictus is an invasive species and the most important vector of arboviruses in temperate regions of the world. A genome sequence is highly helpful for studying the biology of this insect and any improvements of previous assemblies are highly welcomed. For the research community to benefit from any improvement, the assembly has to be made public by posting it to a public repository. It is important, but not sufficient, to provide links to the raw data used in the analysis.
In this ms, a new assembly of the Ae. albopictus genome is described. The new assembly was derived from AalbF2 by anchoring sequencing data with a linkage-map based on 111,328 SNPs derived from RNA-seq data as such the assembly is focused on coding sequences and may have missed repetitive and/or noncoding regions of the genome. The new assembly was used to identify regions of the genome with candidate SNPs associated with photoperiodic diapause.
There are several innovative approaches and data described in the ms, which merits publication. A must before publication is the public release of the new assembly and all the associated analyses that the authors describe, but do not fully disclose, including the comparison of the annotation of the two assemblies (lines 241-254).
The absence of a link to AalbF2B in the current ms prevented a thorough revision of the data.
Additional comments.
- Because the proposed assembly is an improvement of AalbF2 (albeit not in noncoding sequences), I would name it AalbF3 instead of AalbF2B. The name AalbF2B suggests an alternative (A vs B) instead of a progression in improvement (2, 3, …).
- The graphical summary of the work is shown in Figure 1, which I find too wordy. Any reasons for choosing to perform the cross between a tropical male and a temperate female and not the reciprocal cross? Why stopping interbreeding at the 7th generation? Why RNA-seq and not whole genome sequencing was used to identify SNPs? As discussed by the authors, because the linkage-based assembly is based on SNPs identified on cDNAs, maybe part of the 1.09 Gb removed from AalbF2 contains noncoding sequences. I would specify this caveat in the result section, instead of insisting in differences between the two assemblies.
- Page 3, line 113. Given the different evolutionary history Malaysia is part of the native home range of albopictus, Virginia is an invasive area, there may be other differences in tropical and temperate populations apart from diapause. How can you distinguish diapause-related genes and functions from other genes and functions?
- Line 124: Does it simulate a temperate climate (21°C) or a tropical one (humidity is high)? Please motivate the choice for these conditions.
- Lines 156 page 4: it is not clear if reads are single or paired end.
- Lines 159 page 4: some information on the total output of the RNA sequencing would be useful: n° of reads or Gb output.
- Lines 163 page 4: If there is space in the manuscript, I'd add something more here than just forwarding to the repository (eg. which standard bioinformatic tools?).
- Lines 184-195. The analysis is described in great details, which is nice. However, the reasons for the parameters is not explained and is not clear to not experts (ie. Why a LOD score limit between 8-11?).
- Line 230 missing a “to”. Custom R Scripts were used TO obtain…
- Line 260: Why using Bowtie2? It is ok but HISAT2 and STAR (for example) are recognized as the “state of the art” tools to align mRNA-seq on genome. Explain the choice
- Line 285: If I understand correctly, the synteny analysis is based only on genes? Maybe this analysis could have been more powerful integrating scaffolds-vs-chromosome wide comparison. Also, genes may be rearranged in aegypti vs albopictus and some scaffolds may have very few genes reducing the reliability of this type of synteny analysis.
- Line 298: Where was the nix-containing scaffold cut? At which base or in which portion? Here it only says the 5’ was retained on pseudochromosome 1 and the 3’ was moved to ps.chr. 3. Is this info in a supplemental?
- Table 1: It is not clear why contigs have better metrics in AalbF2B but scaffolds do not. It seems like the scaffolds are longer on average, but the biggest ones are smaller than the biggest in AalbF2. Also, is each of the pseudochromosomes continuous, a single sequence?
- Line 431 page 10: the improvement in single-copy genes is great! But how can they explain the small increase in missing and fragmented?
- Line 437: The new assembly is missing 27% of the genes? Was it possible that they were misannotations in AalbF2? Did you check which type of genes they were and if they had a strong support in the AalbF2 annotation? More clearly: how sure you are that these genes have been correctly removed from the assembly?
- Line 448: Did you align paired end reads or single end reads here? In Palatini et al. 2020 (suppl. table S1) they aligned RNAseq to the AalbF2 genome and had 78% of properly paired reads, which is unexpected if a great duplication is present. I think that what they and you see really depends on how sensitive the aligner is and how you set it. Your data is good and indicates more duplication in AalbF2 than in AalbF2B but on a practical point of view it should be noted that a modern gaped aligner like HISAT2 should be able to properly place paired-end reads which are concordantly mapped. This is very different in the case of single-end reads though.
- Figure 4: the figure is very nice, but the resolution could be better.
Author Response
Comment #5: The Asian tiger mosquito Aedes albopictus is an invasive species and the most important vector of arboviruses in temperate regions of the world. A genome sequence is highly helpful for studying the biology of this insect and any improvements of previous assemblies are highly welcomed. For the research community to benefit from any improvement, the assembly has to be made public by posting it to a public repository. It is important, but not sufficient, to provide links to the raw data used in the analysis.
In this ms, a new assembly of the Ae. albopictus genome is described. The new assembly was derived from AalbF2 by anchoring sequencing data with a linkage-map based on 111,328 SNPs derived from RNA-seq data as such the assembly is focused on coding sequences and may have missed repetitive and/or noncoding regions of the genome. The new assembly was used to identify regions of the genome with candidate SNPs associated with photoperiodic diapause.
There are several innovative approaches and data described in the ms, which merits publication. A must before publication is the public release of the new assembly and all the associated analyses that the authors describe, but do not fully disclose, including the comparison of the annotation of the two assemblies (lines 241-254).
The absence of a link to AalbF2B in the current ms prevented a thorough revision of the data.
Response #5: We agree that it is essential that the new assembly is available to the research community in a public repository. The assembly has been deposited at DDBJ/ENA/GenBank under the accession JAFDOQ000000000. The version described in this paper is version JAFDOQ010000000. This information is indicated on lines 435-437. Furthermore, all of the RNAseq reads used to construct the linkage map are now available through GenBank under BioProject accession number PRJNA694122 as indicated on lines 178-179. All scripts used in the analyses for this paper are available in Dryad at https://doi.org/10.5061/dryad.mgqnk98z4 as indicated on line 184. All of this information is also provided in the Data Availability Statement at the end of the manuscript.
Additional comments.
Comment #6: Because the proposed assembly is an improvement of AalbF2 (albeit not in noncoding sequences), I would name it AalbF3 instead of AalbF2B. The name AalbF2B suggests an alternative (A vs B) instead of a progression in improvement (2, 3, …).
Response #6: We have changed AalbF2B to AalbF3 throughout the manuscript and supplementary materials.
Comment #7: The graphical summary of the work is shown in Figure 1, which I find too wordy. Any reasons for choosing to perform the cross between a tropical male and a temperate female and not the reciprocal cross? Why stopping interbreeding at the 7th generation? Why RNA-seq and not whole genome sequencing was used to identify SNPs? As discussed by the authors, because the linkage-based assembly is based on SNPs identified on cDNAs, maybe part of the 1.09 Gb removed from AalbF2 contains noncoding sequences. I would specify this caveat in the result section, instead of insisting in differences between the two assemblies.
Response #7: We have removed text from Figure 1 to make it less wordy and revised the figure legend. As noted below in Response #22, we have submitted high-resolution images of all figures as a separate “.zip” file (Boyle_Figures.zip).
We performed all of the crosses between tropical males and temperate females to maximize the chances of obtaining individual females with a high diapause incidence in the F4 intercross generation for the BSA experiment. The rationale for this decision was that some genes affecting diapause might be sex-linked. To explain this rationale, the following text has been added to lines 143-147: “We performed all of the crosses between tropical males and temperate females to maximize the chances of obtaining individual females with a high diapause incidence in the F4 intercross generation for the BSA experiment. The rationale for this decision was that some genes affecting diapause might be sex-linked.”
To address the issue of why we interbred to the 7th generation, we have added the following text to lines 159-162: “We choose to interbreed to the F7 generation for the linkage map experiment in order to strike a balance between obtaining sufficient recombination to resolve the relative positions of our high-density SNP makers without excessive multiple recombination events between markers.
We also added the following text to lines 155-157 to explain why we choose to interbreed to the F4 generation for the BSA experiment: “We choose to breed to the intercross F4 generation for the BSA experiment because we hypothesized that additional intercross generations would lead to a decrease in the diapause response of intercross females.”
To justify the use of RNAseq, the following text has been added to lines 79-81. “We used RNAseq to identify SNPs because it is an efficient and affordable approach obtain a reduced representation of the genome (i.e., transcribed regions).”
We have added the following text to lines 439-443 (results section) in order to emphasize that our approach of identifying SNPs by RNAseq may have led to the removal of biologically important non-coding sequence from AalbF2: “Of the 1.09 Gb removed from AalbF2, 0.2 Gb were removed as a part of contigs that could not be connected to the linkage maps and it is possible that this sequence contains biologically important non-transcribed elements of the genome; the remaining 0.89 Gb were identified as haplotypes of sequences in the final reassembled genome and were collapsed accordingly”.
We also note that this issue is addressed in the discussion section by the following text on lines 635-639: “Because we used RNAseq-derived SNP data to construct the linkage map used for reassembly, the discarded contigs are likely very gene-poor but may contain biologically important non-transcribed sequences.”
Comment #8: Page 3, line 113. Given the different evolutionary history Malaysia is part of the native home range of albopictus, Virginia is an invasive area, there may be other differences in tropical and temperate populations apart from diapause. How can you distinguish diapause-related genes and functions from other genes and functions?
Response #8: This is an excellent point. We feel that this issue is already addressed by the following text in the original manuscript on lines 113-115: “We identified candidate regions of the Ae. albopictus genome putatively associated with diapause by identifying SNPs that differed both between tropical and temperate samples and between high- and low-diapause bulks.” If this text is not clear, or the reviewers feel that it needs to be more strongly emphasized, we are happy to make a revision. We also note that this issue is partially addressed by our response to Comment # 1 above in which we have changed “associated with diapause” to “putatively associated with diapause throughout the text.
Comment #9: Line 124: Does it simulate a temperate climate (21°C) or a tropical one (humidity is high)? Please motivate the choice for these conditions.
Response #9: We have added the following text to lines 132-134: “These conditions were chosen based on previous optimization of mosquito rearing and to provide a long-day (non-diapause) control for diapause-inducing conditions (8L:16D, 21oC).”
Comment #10: Lines 156 page 4: it is not clear if reads are single or paired end.
Response #10: We have modified the text on lines 175-176 to read as follows: “Paired-end libraries were sequenced on an Illumina HiSeq 4000 instrument...”
Comment #11: Lines 159 page 4: some information on the total output of the RNA sequencing would be useful: n° of reads or Gb output.
Response #11: We have added the following text to line 179: “...(~1.9 billion read pairs, Table S2).” We have added the corresponding information for the TEMP, TROP and BSA analysis on lines 362-363: “...(~ 1.2 billion read pairs, Table S2).”
Comment #12: Lines 163 page 4: If there is space in the manuscript, I'd add something more here than just forwarding to the repository (eg. which standard bioinformatic tools?).
Response #12: We have replaced the reference to “standard bioinformatics tools” with text that specifies the exact software used. The modified text on lines 182-185 is as follows: “…Trimmomatic version 0.39, and SolexaQA++ version 3.1.7.1. Scripts for all analyses are available the Dryad repository located here: https://doi.org/10.5061/dryad.mgqnk98z4. Alignment was performed using STAR version 2.7.1a”
Comment #13: Lines 184-195. The analysis is described in great details, which is nice. However, the reasons for the parameters is not explained and is not clear to not experts (ie. Why a LOD score limit between 8-11?).
Response #13: We have added the following text to lines 210-212 to explain why a LOD score limit between 8-11 was used: “Different LOD scores were used for each group because they contained a different number of SNP markers.”
Comment #14: Line 230 missing a “to”. Custom R Scripts were used TO obtain…
Response #14: Thank you- we have made the change, now on line 253!
Comment #15: Line 260: Why using Bowtie2? It is ok but HISAT2 and STAR (for example) are recognized as the “state of the art” tools to align mRNA-seq on genome. Explain the choice
Response #15: We have added the following text to lines 282-287: “We used the bowtie2 aligner with end-to-end settings for this analysis because it is a commonly used algorithm for short-read alignment that produces easily interpretable results. Although overall alignment rates might be higher using a split-read aligner such as HISAT2 or STAR, because only 11 contigs from the AalbF2 assembly were cut to produce the AalbF3 reassembly (see above), using end-to-end alignments should not affect the comparison of alignment rates between the two assemblies.”
Comment #16: Line 285: If I understand correctly, the synteny analysis is based only on genes? Maybe this analysis could have been more powerful integrating scaffolds-vs-chromosome wide comparison. Also, genes may be rearranged in aegypti vs albopictus and some scaffolds may have very few genes reducing the reliability of this type of synteny analysis.
Response #16: We agree with this point and have added the following text to lines 522-5215: “We note that this analysis is less reliable for assessing local synteny due to possible re-arrangements of gene order between Ae. aegypti vs. Ae. albopictus and the fact that some scaffolds have relatively few genes.”
Comment #17: Line 298: Where was the nix-containing scaffold cut? At which base or in which portion? Here it only says the 5’ was retained on pseudochromosome 1 and the 3’ was moved to ps.chr. 3. Is this info in a supplemental?
Response #17: We have revised the text on lines 320-325 as follows to indicate where the Nix-containing scaffold was cut and moved to Chr. 1: “We therefore moved Nix to its inferred position on pseudochromosome 1 by cutting the scaffold (scaffold 3.129), leaving 2,225,968 bases on the 5’ end (which matched to chromosome 3 of Ae. aegypti) in place, and moving the remaining bases of the 3’ end to pseudochromosome 1, assigning it a position based on the synteny of the two orthologous genes on this (sub-) scaffold with Ae. aegypti.”
Comment #18: Table 1: It is not clear why contigs have better metrics in AalbF2B but scaffolds do not. It seems like the scaffolds are longer on average, but the biggest ones are smaller than the biggest in AalbF2. Also, is each of the pseudochromosomes continuous, a single sequence?
Response #18: We have re-arranged the text on lines 444-451 to more clearly address these issues. The revised text is as follows:
“The contigs that were incorporated into the AalbF3 reassembly were longer than the AalbF2 contigs (7.5 Mb contig N50 compared to 1.2 Mb). In contrast, the scaffolds of the AalbF3 reassembly were smaller than those in the AalbF2 assembly (10 Mb scaffold N50 compared to 56 Mb scaffold N50). This is because we did not join contigs or scaffolds into longer scaffolds if the relative orientation or position was unknown within the linkage map. Concatenating our scaffolds would yield almost full chromosomes but would induce many short scale structural errors.”
Comment #19: Line 431 page 10: the improvement in single-copy genes is great! But how can they explain the small increase in missing and fragmented?
Response #19: We have addressed this issue to adding the following text to lines 469-471: “The small increase in missing and fragmented BUSCO genes is likely because some of the contigs removed during re-assembly contained genes and/or exons”.
Comment #20: Line 437: The new assembly is missing 27% of the genes? Was it possible that they were misannotations in AalbF2? Did you check which type of genes they were and if they had a strong support in the AalbF2 annotation? More clearly: how sure you are that these genes have been correctly removed from the assembly?
Response #20: To address this issue have added the following text to lines 477-480: “The annotated features from AalbF2 that were not included in AalbF3 were likely artifactually duplicated in the AalbF2 assembly, an interpretation supported by the decrease of complete and duplicated BUSCO genes in AalbF3 relative to AalbF2 (Table 2).”
Comment #21: Line 448: Did you align paired end reads or single end reads here? In Palatini et al. 2020 (suppl. table S1) they aligned RNAseq to the AalbF2 genome and had 78% of properly paired reads, which is unexpected if a great duplication is present. I think that what they and you see really depends on how sensitive the aligner is and how you set it. Your data is good and indicates more duplication in AalbF2 than in AalbF2B but on a practical point of view it should be noted that a modern gaped aligner like HISAT2 should be able to properly place paired-end reads which are concordantly mapped. This is very different in the case of single-end reads though.
Response #21: The reads aligned here were paired-end reads. We have modified the text on lines 485-486 as follows to clarify this point: “The overall alignment rate of the paired-end reads to AalbF2 was 81% (±2% standard deviation).”
We agree with the point that different aligners, different settings, and other differences between the reads used by Palatini et al. and the reads used in our experiment make it hard to directly compare the results of our alignments in this section of the manuscript with the results in Table S1 of Palatini et al.
We hope the following text on lines 282-287, included in response to Comment #15 above is sufficient to address this issue: “We used the bowtie2 aligner with end-to-end settings for this analysis because it is a commonly used algorithm for short-read alignment that produces easily interpretable results. Although overall alignment rates might be higher using a split-read aligner such as HISAT2 or STAR, because only 11 contigs from the AalbF2 assembly were cut to produce the AalbF3 reassembly (see above), using end-to-end alignments should not affect the comparison of alignment rates between the two assemblies.”
Comment #22: Figure 4: the figure is very nice, but the resolution could be better.
Response #22: As noted above (Response #7), we have included with our revised submission a zip file with high resolution images for all the figures in the manuscript. (Boyle_Figures.zip)
Reviewer 3 Report
Aedes albopictus, the Asian tiger mosquito, is a major concern for vector control across much of tropical climate due to its role in vectoring several arbovirus borne diseases. Additionally, the species’ adaptability to temperate climate and the ability for some Ae. albopictus strains to undergo photoperiodic diapause, a phenomenon where in unfavorable conditions the eggs do not hatch but remain viable, hinder potential complete elimination of vector by traditional control methods. Authors produced a high density linkage map of Ae. albopictus using ~111K SNPs from two strains- tropical and temperate. This map was subsequently utilized to improve the existing genome assembly AalbF2 (Palatini et al., 2020)– with notable improvements in the expected genome size and percentage of complete single copy BUSCO genes. The authors use this new assembly as a reference to identify regions of genome associated with diapause, a complex environmentally regulated phenomenon present in the temperate but absent in the tropical strain and present a case for this new assembly to be a better reference for Ae. albopictus.
Overall, the manuscript is well written and the comparisons between the new assembly (AalbF2B), the AalbF2 assembly as well as the Aedes aegypti L5 genome assembly (Matthews et al.,2018) emphasize the improvements brought about in AalbF2B. However, I have some concerns regarding the experimental design, particularly since the results have been used to identify genome regions associated with diapause.
Major comments
Line 102- crossing scheme- For ascertaining the loci pertaining to diapause, a single tropical male was crossed with a single temperate female. What was the genetic background for these individuals? Were the lab populations for each inbred for ~5 generations to produce a more uniform background for parental cross? Why was a reciprocal cross between tropical female X temperate male not performed? In order to isolate the regions involved in diapause, the two F4 lines with total of 4 groups of high and low diapause were evaluated and the power of recombination between the two strains was used. However, is this a true comparison when the F0 females are only taken from the temperate pool? An additional experiment with reciprocal cross should be performed (temperate male X tropical females) to obtain a true representation of non diapause female groups and exclude any SNPs which would be expressed in such groups compared to high diapause incidence group. A comparison of such females to high DI females should in fact, provide a clearer peak for BSA analysis. In a reciprocal cross containing tropical females X temperate males with exact experiment set up otherwise, the SNPs expressed would be a better pool to compare with.
Line 196-200- ‘heterozygous grandparents led to removal of linkage group 10-12’. Another reason why grandparents’ genetic background should have been homogenized first by performing a few rounds of inbreeding.
Lines 233-237- talks about both grandparent F0 being a heterozygote and also long stretches of inbred segments? Please clarify.
Minor comments
Line 124- “TEMP colony was reared at 21oC, at 16L:8D photoperiod..” Should the dark period (D) not be more for inducing diapause? Eg. 8L:16D or 10L:14D?
Lines 125-127- “the TROP line was maintained under conditions similar to TEMP (21oC)..” Ae. albopictus tropical strains are typically reared in lab at 27oC, a significant temperature rearing change may impact the expression of diapause and therefore the experiment design.
Line 316,317- “Bulk seg analysis..” should be in italics and accordingly spaced.
Author Response
REVIEWER #3:
Major comments
Comment #23: Line 102- crossing scheme- For ascertaining the loci pertaining to diapause, a single tropical male was crossed with a single temperate female. What was the genetic background for these individuals? Were the lab populations for each inbred for ~5 generations to produce a more uniform background for parental cross? Why was a reciprocal cross between tropical female X temperate male not performed? In order to isolate the regions involved in diapause, the two F4 lines with total of 4 groups of high and low diapause were evaluated and the power of recombination between the two strains was used. However, is this a true comparison when the F0 females are only taken from the temperate pool? An additional experiment with reciprocal cross should be performed (temperate male X tropical females) to obtain a true representation of non diapause female groups and exclude any SNPs which would be expressed in such groups compared to high diapause incidence group. A comparison of such females to high DI females should in fact, provide a clearer peak for BSA analysis. In a reciprocal cross containing tropical females X temperate males with exact experiment set up otherwise, the SNPs expressed would be a better pool to compare with.
Response #23: The lab populations were not inbred before performing the parental cross. To clarify this point, we have added the following text to lines 136-138: “Both colonies were maintained as large (census size >300 individuals), outbred populations before initiating the intercross matings described below.”
We agree that it would have been advantageous for the BSA analysis to perform the reciprocal TROP-female X TEMP-male cross. However, because this was our first crossing experiment, we choose the more conservative strategy of performing all crosses with TEMP-females x TROP-males to try and ensure we obtained a sufficient expression of diapause in our intercross lines. This rationale is explained in response to Comment #7 above and repeated here for convenience (the text has been added to lines 143-147 of the manuscript).
“We performed all of the crosses between tropical males and temperate females to maximize the chances of obtaining individual females with a high diapause incidence in the F4 intercross generation for the BSA experiment. The rationale for this decision was that some genes affecting diapause might be sex-linked.”
Comment #24: Line 196-200- ‘heterozygous grandparents led to removal of linkage group 10-12’. Another reason why grandparents’ genetic background should have been homogenized first by performing a few rounds of inbreeding.
Response #24: Yes, we agree. In retrospect it might have been advantageous to perform inbreeding before the intercrossing. We have not added a statement to this effect to the current version of the manuscript, but would be happy to do so if Reviewer #3 feels that this is important.
Comment #25: Lines 233-237- talks about both grandparent F0 being a heterozygote and also long stretches of inbred segments? Please clarify.
Response #25: The inference regarding long stretches of inbred segments is speculative and has therefore been removed. The text on the previous line (“...seemed to be caused by gaps without markers...”) is the important issue. There may be multiple reasons why these gaps without markers occurred.
Minor comments
Comment #26: Line 124- “TEMP colony was reared at 21oC, at 16L:8D photoperiod..” Should the dark period (D) not be more for inducing diapause? Eg. 8L:16D or 10L:14D?
Response #26: Here we are describing the non-diapause rearing conditions used for colony maintenance. This issue was also raised by Reviewer #2 (see Comment #9 above). We have added the following text to lines 132-134 to clarify this issue: “These conditions were chosen based on previous optimization of mosquito rearing and to provide a long-day (non-diapause) control for diapause-inducing conditions (8L:16D, 21oC).”
Comment #27: Lines 125-127- “the TROP line was maintained under conditions similar to TEMP (21oC)..” Ae. albopictus tropical strains are typically reared in lab at 27oC, a significant temperature rearing change may impact the expression of diapause and therefore the experiment design.
Response #27: Note that as indicted on lines 99-100 “...tropical populations of Ae. albopictus are genetically incapable of diapause...”
Also, we hope that this issue is addressed by our Response #26 above. We have also modified the text on lines 330-333 of the revised manuscript in the “Bulk segregant analysis (BSA): measuring diapause phenotypes” section to further clarify this issue: “Here, we define the diapause phenotype of an individual female as diapause incidence (DI), the proportion of diapause eggs produced by a female maintained under an unambiguous short-day photoperiod (8L:16D) at 21oC, conditions that produce the optimal expression of diapause in Ae. albopictus [25, 37]. “
Comment #28: Line 316,317- “Bulk seg analysis..” should be in italics and accordingly spaced.
Response #28: Thank you- we have made this correction on lines 346-347 of the revised manuscript.
Round 2
Reviewer 3 Report
I have read the responses to my comments. The changes in the text provided by the authors are sufficient to merit acceptance of the paper. in the current format.